# Temperature-Driven Synthesis of 1D Fe₂O₃@3D Graphene Composite Applies as Anode of Lithium-Ion Batteries

Shengyuan Zhu [1], Ruizhi Li [1,*], Jiapeng Xu [2], Liu Yang [1] and Yingke Zhou [1,*]

1 The State Key Laboratory of Refractories and Metallurgy, College of Materials and Metallurgy, Wuhan University of Science and Technology, Wuhan 430081, China; aprildayzsy@outlook.com (S.Z.); koyukikirisame@163.com (L.Y.)
2 State Key Laboratory of Electronic Thin Films and Integrated Devices, University of Electronic Science and Technology of China, Chengdu 610054, China; xu7639@gmail.com
* Correspondence: rzli@wust.edu.cn (R.L.); zhouyk@wust.edu.cn (Y.Z.)

**Abstract:** A series of Fe₂O₃-anchored three-dimensional graphene (3DG) composites are synthesized via hydrothermal and annealing methods. The Fe₂O₃ nanocrystals in composites display nanocubes, one-dimensional (1D) nanorods and ellipsoids at hydrothermal temperatures of 120 °C, 150 °C and 180 °C, respectively. Notably, the composite synthesized at 150 °C shows 1D Fe₂O₃ uniformly embedded in 3DG, forming an interpenetrating 1D-3D (three-dimensional) structure. This combined structure is beneficial in improving the electrochemical stability and accelerating the Li⁺ diffusion rate. When used as anode for lithium-ion batteries (LIBs), the optimized 1D-3D Fe₂O₃@3DG composite delivers a reversible specific capacity of 1041 mAh g⁻¹ at 0.1 A g⁻¹ and maintains a high reversible specific capacity of 775 mAh g⁻¹ after 200 cycles. The superior electrochemical properties of Fe₂O₃@3DG are a result of the stable interpenetrate structure, enhanced conductivity, and buffered volume change. These results suggest that Fe₂O₃@3DG composites have significant potential as advanced anode materials for LIBs and the combined 1D-3D structure also provides inspiration for other electrode material structure design.

**Keywords:** Fe₂O₃; graphene; lithium-ion battery; composite; anode

## 1. Introduction

To meet the increased demand for consumer electronics, electric automobiles, and high-performance energy storage equipment, LIBs are widely applied for their superior energy density, long service life, and environmental friendliness [1,2]. Nevertheless, graphite as an anode material applied in conventional commercial LIBs cannot satisfy social demands on account of its undesirable theoretical capacity (372 mAh g⁻¹) and adverse lithiation potential, which increases the tendency for formation of lithium dendrites during fast charging [3–5]. These challenges have prompted a desire to explore using high-performance anodic substitutes to improve LIB electrochemical performance.

Transition metal oxides (TMOs) have been recognized as prospective anode materials thanks to their reversible conversion reaction causing an excellent theoretical storage capacity (500–1000 mAh g⁻¹) [6]. Among TMOs, Fe₂O₃ has a high theoretical capacity (1007 mAh g⁻¹) and a safe lithiation potential (about 1 V vs. Li/Li⁺), making it a potential substitute for high-performance anode material [7–9]. However, its large volume expansion and low electron transport rate leads to quick capacity loss and inferior rate performance in long-term operation [10–12]. Recently, it was discovered that electrochemical properties of Fe-based oxide anode can be enhanced when hybridized with graphene. For example, Oliva et al. [13] prepared a CoFe oxide/graphene nanocomposite using the wet-chemistry method, which obtained an optimized electrochemical performance. Li et al. [14] synthesized graphene-Fe₂O₃ hybrid sheet materials by the one-step hydrothermal method and delivered a retention capacity of 765 mAh g⁻¹ after 100 cycles at 0.2 A g⁻¹. These

composites have a shortened diffusion path for electron/ion transfer and improved $Li^+$ storage capacity, but they still suffer from the agglomeration of $Fe_2O_3$ crystals, resulting in capacity degradation during long-term cycling. Moreover, the nanostructure formed by $Fe_2O_3$ and graphene cannot remain stable and greatly increases the risk of structural collapse, giving rise to a poor Coulombic efficiency. Therefore, it is crucial to design a $Fe_2O_3$@graphene composite anode with a stable nanostructure and uniform dispersion.

Recently, hydrothermal method can be applied to effectively disperse GO (graphene oxide) layers and make them self-assemble into an ordered 3D structure by utilizing their hydrophilic and electrostatic repulsive effects [15,16]. Its interconnecting 3D structure can restrict the volume fluctuations, and the high conductive network can enhance the kinetic behavior of the anode [17,18]. Meanwhile, the 1D nanostructures are known to exhibit unique electronic properties due to their quantum confinement effects. These properties allow 1D-$Fe_2O_3$ to form transport paths for electrons and ions in specific directions, which reduces polarization and increases rate capabilities [19]. Thus, we propose the design of a 1D-3D interpenetrating structure by combining 1D-$Fe_2O_3$ and 3DG. The 1D nanostructure can easily pierce into 3DG framework to form a highly stable structure and restraining the agglomeration of $Fe_2O_3$ crystals. The 1D-3D structure also allows accommodation of the strain caused by $Li^+$ insertion/removal in a specific direction in the presence of abnormal expansion, as well as providing a larger contact surface between electrolyte and active material, thus increasing the electrochemical reaction rate [10,20–22]. In addition, research has shown that $SO_4^{2-}$ and $Cl^-$ have a structural guiding effect on $Fe_2O_3$ formation during the hydrothermal process, fabricating various 3D (nanocubes [23], 3D urchin-like [24]) and 1D (nanorods [25]) morphologies. However, hydrothermal crystallization is a complex process coupled by various factors such as solution concentration, temperature, pressure and so on. Achieving control over the crystal morphology can be a challenging task. Therefore, the main focus of this work is to explore a facile approach to obtain controlled crystal morphology and prepare an $Fe_2O_3$@3DG hybrid with 1D-3D interpenetrating structure.

In this work, we report the synthesis of $Fe_2O_3$@3DG composite by hydrothermal and annealing methods. We investigated how temperature variables impact crystallographic morphology by using the strong correlation between crystallization and temperature during hydrothermal synthesis. A series of $Fe_2O_3$@3DG composites with varying morphologies including nanocubes, 1D-nanorods and ellipsoidal crystals was achieved. Specifically, the composite prepared at 150 °C exhibited a unique 1D-3D interpenetrating structure. When applied as anodes for LIBs, the synthesized 1D-$Fe_2O_3$@3DG composite delivered a remarkable capacity and improved cycling stability. These results are ascribed to the unique interpenetrating nanostructure with excellent stability, which alleviates the volume expansion of $Fe_2O_3$, enlarges the contact surface of electrolyte and electrode, and promotes the rapid transport of $Li^+$ and electron. These results prove the prospective potential of 1D-$Fe_2O_3$@3DG composite as an effective anode material for LIBs.

## 2. Results and Discussion

### 2.1. Material Characterization

The preparation procedure for the $Fe_2O_3$@3DG is displayed in Figure 1. Firstly, the mixture is transferred to an oven for hydrothermal rection. When the temperature of the autoclave increases, graphene layers highly interconnected are reduced to a 3DG framework, as shown in Figure 2d–f. The $Fe^{3+}$ reacts with $OH^-$ and $O_2$ to form a FeOOH solution. The formed FeOOH nuclei attach to the surface of 3DG through hydrolysis of $FeCl_3$, and subsequently grow into crystals of a specific size and shape. During final annealing in Ar, all FeOOH phases are completely converted to $Fe_2O_3$. Related studies have shown that both $Cl^-$ and $SO_4^{2-}$ have a strong coordination property for $Fe^{3+}$, leading to the aggregation of fine primary particles [22,24].

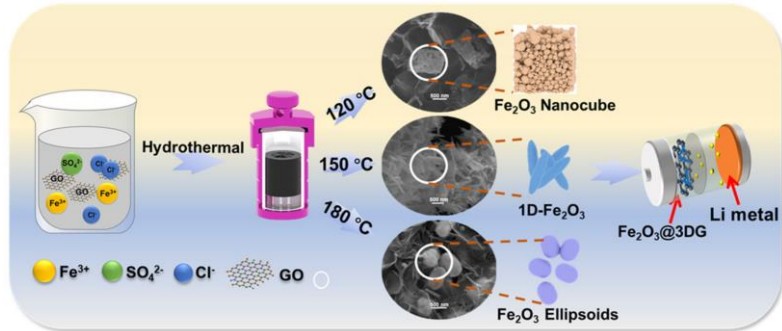

**Figure 1.** Formation process of $Fe_2O_3$@3DG composites.

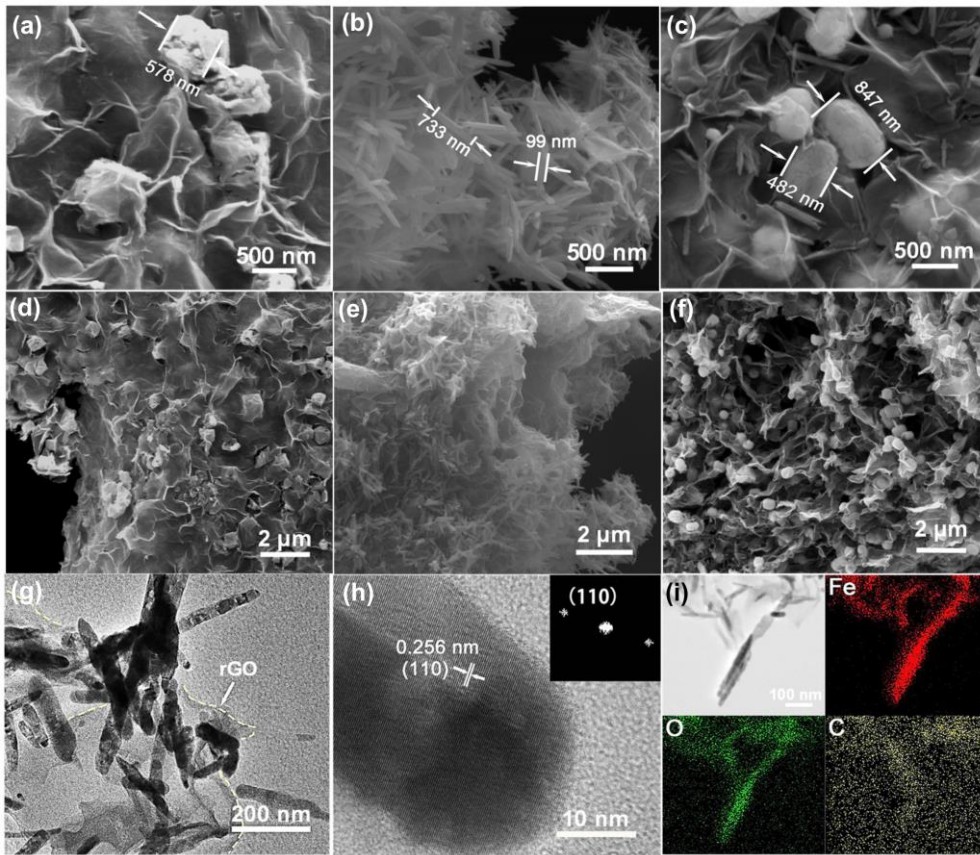

**Figure 2.** SEM images of (**a**,**d**) $Fe_2O_3$@3DG-120, (**b**,**e**) $Fe_2O_3$@3DG-150, (**c**,**f**) $Fe_2O_3$@3DG-180. (**g**) TEM image of $Fe_2O_3$@3DG-150, (**h**) HRTEM image of $Fe_2O_3$@3DG-150. Inset in (**h**) is SAED pattern. (**i**) TEM of $Fe_2O_3$@3DG-150 and elemental mapping results of Fe, O, and C.

When the hydrothermal temperature is 120 °C, $Cl^-$ dominated the coordination and bonded with FeOOH, causing the fine primary particles to aggregate to form nanocubes [23,26]. In Figure 2a, the SEM image shows details of the nanocubes with a side length of about 578 nm. The nanocubes of $Fe_2O_3$ are uniformly distributed and well attached to the 3DG framework, as depicted in Figure 2d. However, when the temperature increases to 150 °C, $SO_4^{2-}$ plays an important role as a ligand for $Fe^{3+}$ and is absorbed on the tail end facets of the c-axis of the FeOOH nucleus through a monodentate structure (Fe-O-$SO_3$), resulting in 1D-FeOOH nanorods [24]. As demonstrated in Figure 2b, it can be noticed that the 1D-structures have a diameter of ~99 nm and a length of ~733 nm. The graphene layer forms a cross-linked framework and a large amount of 1D-$Fe_2O_3$ pierces the structure of 3DG to hold tightly, forming a unique 1D-3D interpenetrating structure (Figure 2e). Finally, as the reaction temperature rises to 180 °C, the crystal growth is accelerated and the tips of

the nanorods begin to dissolve towards the interior along the c axis, forming ellipsoidal crystals with the coordination of $Cl^-$ and $SO_4^{2-}$ [27]. The crystals are approximately 847 nm in length and 482 nm in diameter (Figure 2c). Figure 2f displays the morphology of the ellipsoidal crystals that are uniformly embedded in a 3DG layers. All results indicate that well-hybridized $Fe_2O_3$@3DG composites have been successfully prepared.

The low-magnification TEM image displayed in Figure 2g exhibits the 1D morphology of $Fe_2O_3$ embedded in reduced graphene oxide (rGO) layers. Meanwhile, as presented in the high-resolution TEM images of $Fe_2O_3$@3DG-150 (Figure 2h), the interplanar distance of lattice fringes are measured to be 0.256 nm, corresponding to the spacing of the (110) crystal plane of $Fe_2O_3$ [28,29]. Moreover, it is noteworthy that the selected area electron diffraction (SAED) image (inset of Figure 2h) exhibits a pair of symmetrical diffracted bright spots of 1D-$Fe_2O_3$, indicating its single crystal properties and corresponding to the (110) crystal plane of $Fe_2O_3$ [30]. The growth of $Fe_2O_3$ single crystals in a specific direction is ascribed to the directional coordination of $SO_4^{2-}$ at the appropriate temperature (150 °C) [24]. Furthermore, the EDX mappings results (Figure 2i) display the uniform distribution of Fe and O elements in 1D $Fe_2O_3$, and the C element has a wider distribution range due to the coverage of the thin graphene layer. These results confirm that 1D-$Fe_2O_3$ is uniformly encapsulated in the graphene layers.

TGA was performed to identify the graphene content in the $Fe_2O_3$@3DG-150 composite. As shown in Figure 3, the weight percentages of 3DG in the $Fe_2O_3$@3DG-120, $Fe_2O_3$@3DG-150, $Fe_2O_3$@3DG-180 are 42.18 wt%, 39.76 wt% and 21.31 wt%. To further investigate the properties of composites, $Fe_2O_3$@3DG-120, $Fe_2O_3$@3DG-150, and $Fe_2O_3$@3DG-180 were dissolved in alcohol by ultrasonic assistance to make suspensions, and their precipitation results after 24 h of standing are shown in the inset optical photograph. Only the $Fe_2O_3$@3DG-150 sample is completely precipitated, and both $Fe_2O_3$@3DG-120 and $Fe_2O_3$@3DG-180 samples have apparent black suspension, which indicates an excellent mechanical anchor between 1D-$Fe_2O_3$ and 3DG in the $Fe_2O_3$@3DG-150. The $Fe_2O_3$@3DG-120 has no clarification due to the lower content of the $Fe_2O_3$ in the composite. The bottom of sample $Fe_2O_3$@3DG-180 shows obvious reddish solid precipitation, while the upper solution remains black, which indicates that the $Fe_2O_3$ particles are detached from 3DG in the sonication procedure. This phenomenon indicates a lack of interaction between large ellipsoidal $Fe_2O_3$ particles and the 3DG, causing easy detaching during ultrasonic process.

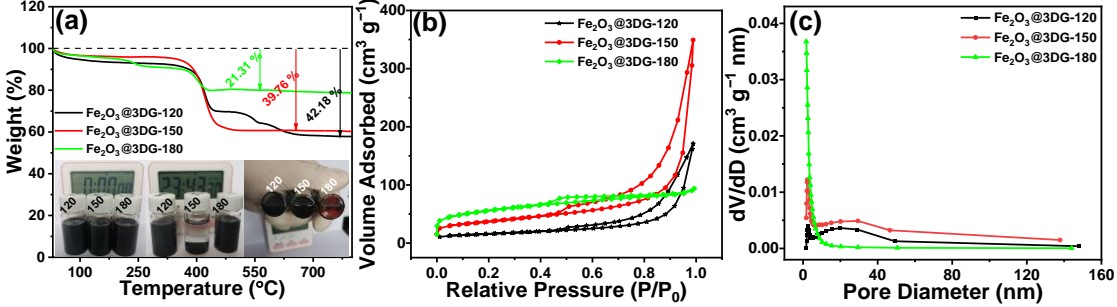

**Figure 3.** (**a**) TGA curves of $Fe_2O_3$@3DG120, $Fe_2O_3$@3DG-150 and $Fe_2O_3$@3DG-180. Inset shows the optical photos of the precipitation results after 24 h. (**b**) $N_2$ adsorption/desorption isotherms and (**c**) pore size distribution of $Fe_2O_3$@3DG-120, $Fe_2O_3$@3DG-150 and $Fe_2O_3$@3DG-180.

$N_2$ adsorption/desorption isotherms are shown in Figure 3b. According to the IUPAC classification, the adsorption-desorption isotherms of all samples exhibited a typical IV isotherm pattern [31]. As for the all composites, the hysteresis loop mainly occurs at a wide relatively pressure ($P/P_0$) ranging of 0.4 to 1.0. The $Fe_2O_3$@3DG-180 has the largest Brunauer–Emmett–Teller (BET) surface area of 196.6 $m^2$ $g^{-1}$. Based on the pore size distribution curves (Figure 3c), it can be speculated that the majority of the pore sizes are distributed within the diameter range of <5 nm, indicating that the pores in $Fe_2O_3$@3DG-180 are mainly results from void spaces formed by the stacking of a lot of $Fe_2O_3$ crystals

(78.69 wt%). The BET surface areas are 54.4 and 129.4 $m^2$ $g^{-1}$ for $Fe_2O_3$@3DG-120 and $Fe_2O_3$@3DG-150, respectively. Both composites exhibit similar graded porous features consisting of pores with diameters of about 2.5 nm and mesoporous with diameters of 20–30 nm. $Fe_2O_3$@3DG-150 has a larger specific surface area than $Fe_2O_3$@3DG-120, which is attributed to the high length–diameter ratio of 1D-$Fe_2O_3$ piercing into the 3DG framework to form more pores.

X-ray diffraction (XRD) further investigates the crystallographic texture and components of the obtained composites. As depicted in Figure 4a, all peaks of $Fe_2O_3$@3DG composites and pure $Fe_2O_3$ ($Fe_2O_3$-150) in the pattern can be perfectly assigned to the face-centered cubic $Fe_2O_3$ phase (JCPDS No. 33-0664) [3]. Meanwhile, almost the same diffraction features appear in the diffraction pattern of the $Fe_2O_3$-150 sample, implying the $Fe_2O_3$ component in $Fe_2O_3$@3DG-150 is not reduced during the final annealing process. The highest diffraction peak at 35.61° indicates that it grows oriented along the (110) plane, such results are consistent with the TEM results. The same peak location characteristics are also found in $Fe_2O_3$@3DG-120 and $Fe_2O_3$@3DG-180, indicating the presence of the $Fe_2O_3$ phase in the composite. However, the peaks at 35.61° in $Fe_2O_3$@3DG-120 and $Fe_2O_3$@3DG-180 no longer exhibit the highest diffraction intensities, but are replaced by the peak with the highest intensity at 33.0°, as depicted in the inset. This is attributed to the anisotropy of the $Fe_2O_3$ crystals, which is a characteristic of bulk crystals. In addition, the small and broad diffraction (002) peak near 26° in all $Fe_2O_3$@3DG composites indicates the presence of disordered 3DG in the sample [9].

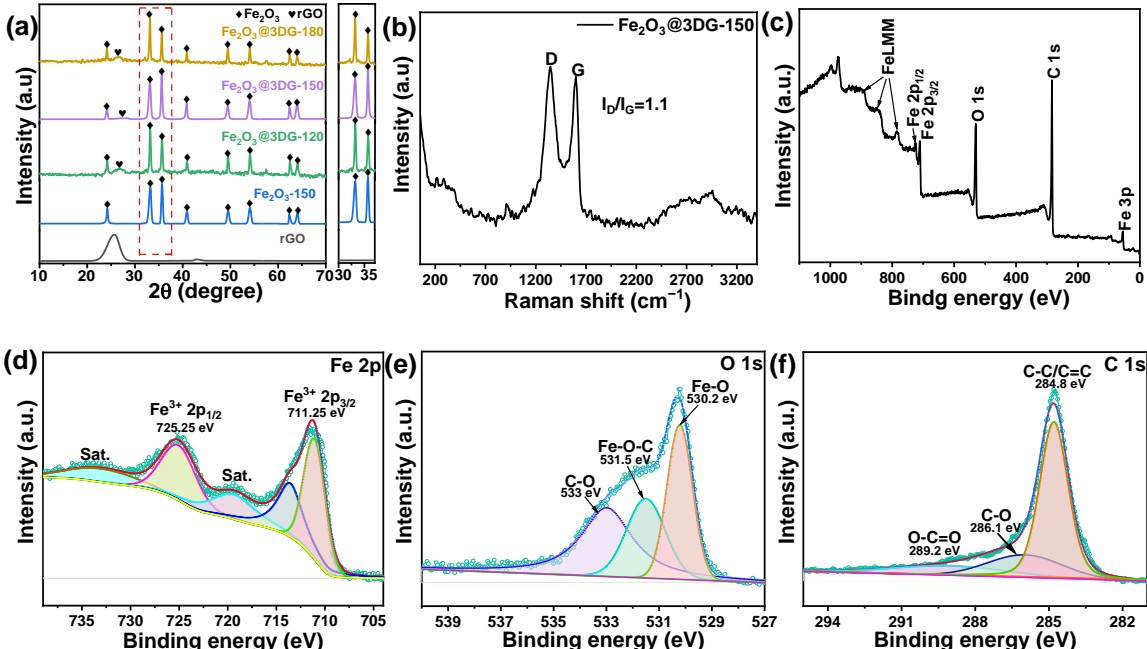

**Figure 4.** (**a**) XRD patterns of $Fe_2O_3$@3DG-120, $Fe_2O_3$@3DG-150, $Fe_2O_3$@3DG-180, $Fe_2O_3$-150 and rGO. Inset is the intensity comparison patterns in the range of 30 to 37°. (**b**) Raman spectrum of $Fe_2O_3$@3DG-150. (**c**) XPS full spectrum of $Fe_2O_3$@3DG-150. XPS spectrum for (**d**) Fe 2p, (**e**) O 1s and (**f**) C 1s.

Raman spectra of the sample were carried out to identify the existence of carbon. As shown in Figure 4b, two strong bands near 1343.8 $cm^{-1}$ and 1600.3 $cm^{-1}$ related to unordered structural defects generate in graphite (D band) and ordered graphite $sp^2$ tensile mode (G band), respectively. The D band represents the disorder degree of 3DG, which indicates the degree of defects and mismatches, and the G band represents the tensile vibration of the C–C bond, indicating the degree of graphitization of 3DG [29,32]. The $I_D/I_G$ value in the composite is about 1.1. This distinction in the $I_D/I_G$ ratio indicates

the reduction of GO results in an increase in the quantity of $sp^2$ domains in $Fe_2O_3$@3DG-150 [18,33,34].

In order to evaluate the chemical composition and valence state of the samples, XPS was implemented. Figure 4c displays the XPS survey spectrum of $Fe_2O_3$@3DG-150. Evident peaks of Fe 2p, O1s, and C 1s are observed in the composite, and no other elemental peak is discovered, which reveals the $Fe_2O_3$@3DG-150 mainly composes of three elements of Fe, O, and C. As revealed in the fitted spectra in Figure 4d, the several evident peaks of $Fe_2O_3$ at 711.15 eV, 725.25 eV and 719.4 eV and 733 eV are related to Fe $2p_{3/2}$, Fe $2p_{1/2}$ of $Fe^{3+}$ and their satellite peak, respectively [34]. The peak at 713.6 eV is assigned to the C-O-Fe bond, indicating that the linkage of anchored $Fe_2O_3$ to 3DG [35]. In the O 1s spectrum (Figure 4e), three distinct peaks at 530.2, 531.5, and 533.0 eV correspond to Fe-O, Fe-O-C and C-O, respectively [36]. Significantly, the Fe-O-C bond is considered to significantly contribute to maintaining a complete and accordant anchoring nanostructure, and promoting the contact of $Fe_2O_3$ and 3DG via chemical bond between 1D-$Fe_2O_3$ and 3DG [37]. This is consistent with the results of the dispersion-precipitation experiment. The XPS spectrum of C 1s is fitted by three peaks (Figure 3f), in which the peak at 286.1 eV corresponds to the C-O bond, the peak at 284.8 eV relates to C-C/C=C, and the peak at 289.2 eV attributes to O-C=O, respectively [35,38]. The highest intensity C-C bond implies that GO is effectively reduced during hydrothermal process.

## 2.2. Electrochemical Characterization

Figure 5a shows the first five CV curves of the $Fe_2O_3$@3DG-150 at a scan rate of 0.1 mV $s^{-1}$ in the voltage window of 0.01–3.0 V. In the initial cathodic process, the presence of the peak at 1.55 V indicates that the irreversible conversion between $Li^+$ and $Fe_2O_3$ leads to the formation of a solid electrolyte-interphase (SEI) layer [39]. Another apparent characteristic peak at 0.65 V was attributed to the formation of irreversible $Li_xFe_2O_3$ ($Fe_2O_3 + Li^+ + e^- \rightarrow Li_xFe_2O_3$) [40]. In addition, subsequent CV curves almost overlap in the following cycles, indicating superior reversibility. The recurring peak centered at 0.83 V in the subsequent cycles indicates the reversible redox process, which can be described by the following equations [41]:

$$Fe^{3+} \text{ to } Fe^{2+}: Fe_2O_3 + 6Li^+ + 6e^- \rightarrow 2FeO + 3Li_2O \tag{1}$$

$$Fe^{2+} \text{ to } Fe^0: FeO + 2Li^+ + 2e^- \rightarrow Li_2O + Fe \tag{2}$$

During the following cycles, a pair of wide but unclear oxidation peaks centered at 1.63 V and 1.86 V correspond to the conversion of $Fe^0$ to $Fe^{2+}$ and $Fe^{3+}$, which are the inverse process of Equations (1) and (2) [35]. Furthermore, the nearly overlapping CV curves in the successive cycles indicate that the 1D-3D interpenetrating nanostructure of $Fe_2O_3$@3DG-150 possesses excellent cyclability and reversibility during charging–discharging.

Figure 5b represents the discharge/charge curves of $Fe_2O_3$@3DG-150 for the first five cycles at 0.1 A $g^{-1}$, where the voltage plateaus are consistent with the peaks of the CV curves. The composite delivers an initial discharge/charge specific capacity of 1646/1070 mAh $g^{-1}$ and the initial Coulombic efficiency is 65%, which is attributed to the generation of SEI film and the irreversible reaction of lithium ions [18]. However, the discharge capacity almost equals the charge capacity in subsequent cycles implying the great invertibility of the electrode. This is attributed to the 1D-3D hybrid structure that inhibits $Fe_2O_3$ agglomeration and alleviates the volume expansion effect, thus retaining the capacity.

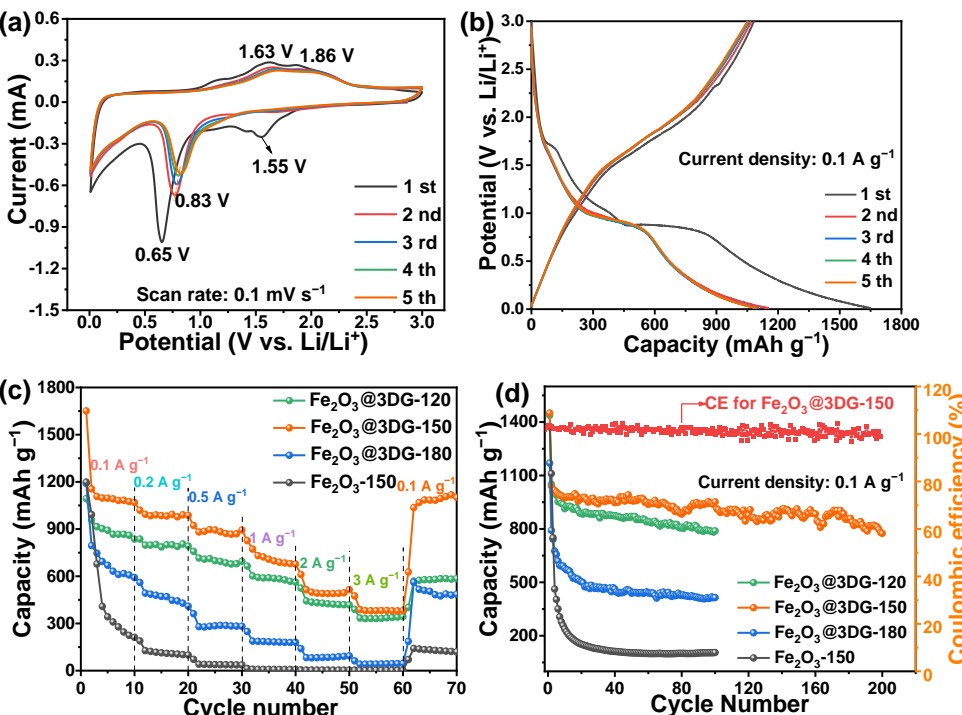

**Figure 5.** (**a**) The first five CV curves of the Fe$_2$O$_3$@3DG-150 at a scan rate of 0.1 mV s$^{-1}$. (**b**) The first five discharge and charge curves of Fe$_2$O$_3$@3DG-150 at a current density of 0.1 A g$^{-1}$. (**c**) Rate performance of various Fe$_2$O$_3$@3DG at different current densities. (**d**) Cycling performance of various Fe$_2$O$_3$@3DG at 0.1 A g$^{-1}$.

Furthermore, the rate properties of the Fe$_2$O$_3$@3DG anodes were performed at different current densities from 0.1 to 3 A g$^{-1}$, as depicted in Figure 5c. Obviously, all composite anodes exhibit higher discharge capacities than pure Fe$_2$O$_3$ electrodes (Fe$_2$O$_3$-150) at each current density. This indicates an excellent synergy between Fe$_2$O$_3$ and 3DG, resulting in abundant active sites for Li$^+$ storage. The reversible discharge capacities of Fe$_2$O$_3$@3DG-150 are 1069, 987, 893, 677, 515 and 381 mAh g$^{-1}$ at a current density of 0.1, 0.2, 0.5, 1, 2 and 3 A g$^{-1}$, respectively. Significantly, a high specific capacity of 1101 mAh g$^{-1}$ is renewed after the current density reverts to 0.1 A g$^{-1}$, indicating the 1D-3D structure has a superb stability and excellent reversibility. The Fe$_2$O$_3$@3DG-120 has discharge capacities of 836, 792, 695, 562, 418 and 343 mAh g$^{-1}$ at 0.1, 0.2, 0.5, 1, 2 and 3 A g$^{-1}$, respectively. In comparison, the Fe$_2$O$_3$@3DG-180 has capacities of 591, 409, 282, 180, 93, and 45 mAh g$^{-1}$ at the same current densities. These results are worse than Fe$_2$O$_3$@3DG-150 owing to the smaller proportion of Fe$_2$O$_3$ in the former (57.8 wt%) than the latter (60.3 wt%). The poorer rate performance of Fe$_2$O$_3$@3DG-180 is due to the stacking of a lot of Fe$_2$O$_3$ crystals and the majority of micropores are not beneficial for electrolyte permeation. Meanwhile, in Fe$_2$O$_3$@3DG-120 and Fe$_2$O$_3$@3DG-180, a large percentage of the Li$^+$ storage sites are limited to the surface of the bulk Fe$_2$O$_3$ (nanocubes and ellipsoidal crystals), which are not favorable for Li$^+$ reaction [42] However, in the Fe$_2$O$_3$@3DG-150, due to the large surface area created by the high length–diameter ratio of 1D-Fe$_2$O$_3$ into the 3DG framework to form a graded porous structure, an enhanced pseudo-capacitive effect can be achieved, leading to higher lithium storage.

In order to explore the electrochemical stability of composite anodes. The results of cycling performance measured at 0.1 A g$^{-1}$ are shown in Figure 5d. The composite electrodes display better cycling performance than the pure Fe$_2$O$_3$ electrode, further demonstrating the positive impact of 3DG on enhancing the electrode performance. After the first charge, Fe$_2$O$_3$@3DG-150 has a reversible discharge specific capacity of 1041 mAh g$^{-1}$, and a high specific capacity of 952 mAh g$^{-1}$ after 100 cycles. A discharge specific capacity of 775 mAh g$^{-1}$ is maintained even after 200 cycles that is about 74.5% of the capacity

retention rate. A near 100% Coulombic efficiency and much higher reversible capacity than $Fe_2O_3$-150 during cycling indicate that the utilization of 3DG can establish an efficient conductive network and mitigate the impact caused by the volume fluctuations of $Fe_2O_3$ during the lithiation/delithiation process, which improves cycling stability. The cycling performance of $Fe_2O_3$@3DG-150 is better than many previous similar works, summarized in Table 1 [11,14,34,43–49]. However, the $Fe_2O_3$@3DG-120 and $Fe_2O_3$@3DG-180 demonstrate lower specific capacity of 789 and 414 mAh $g^{-1}$ after 100 cycles at 0.1 A $g^{-1}$, while maintaining capacity retention rates of 75.0% and 52.3%, respectively. The reasons for this phenomenon can be attributed to following factors: (I) The bulk $Fe_2O_3$ in $Fe_2O_3$@3DG-120 and $Fe_2O_3$@3DG-180 is easily dislodged from the graphene sheets due to lithiation expansion, and even the complex mesh of 3DG is not well maintained in terms of structural stability. (II) The 1D structure $Fe_2O_3$ can easily pierce 3DG to form a stable 1D-3D interpenetrating structure. This interpenetrating structure possesses graded porous features that provide unparalleled structural stability and capacity reversibility while creating a large surface area to provide abundant lithium storage sites. (III) The bulk $Fe_2O_3$ (nanocubes and ellipsoidal crystals) are accompanied by faradaic charge storage with crystallographic phase change leading to structural disruption. However, the high length–diameter ratio of 1D $Fe_2O_3$ allows for reversible faradaic charge storage due to its large surface and near-surface.

**Table 1.** The comparison of LIBs anode performances between this work and the previous similar composites.

| Anode Material | Current Density | Initial Capacity (mAh $g^{-1}$) | Cycle Number | Reversible Capacity (mAh $g^{-1}$) | Ref |
|---|---|---|---|---|---|
| Pomegranate-shaped $Fe_2O_3$/C | 0.1 A $g^{-1}$ | ~900 | 100 | 705 | [11] |
| graphene-$Fe_2O_3$ | 0.2 A $g^{-1}$ | 1466 | 100 | 765 | [14] |
| $\alpha$-$Fe_2O_3$@graphene | 0.1 A $g^{-1}$ | ~1300 | 100 | 745 | [34] |
| graphene/$\alpha$-$Fe_2O_3$ | 0.1 A $g^{-1}$ | ~1200 | 100 | 607 | [43] |
| N-$Fe_2O_3$@Carbon | 0.1 C | 1049 | 30 | 800 | [44] |
| $Fe_2O_3$@graphite | 0.1 A $g^{-1}$ | ~1200 | 100 | 865 | [45] |
| $Fe_2O_3$/rGO/CNFs | 0.1 A $g^{-1}$ | 1378 | 150 | 811 | [46] |
| $\alpha$-$Fe_2O_3$@rGO | 0.1 A $g^{-1}$ | 1268 | 100 | 781 | [47] |
| $Fe_2O_3$/graphene | 0.5 A $g^{-1}$ | 1086.3 | 100 | 653.2 | [48] |
| $Fe_3O_4$/graphene | 0.1 A $g^{-1}$ | 1462 | 100 | 985 | [49] |
| 1D-$Fe_2O_3$@3DG | 0.1 A $g^{-1}$ | 1451.1 | 100 | 951.9 | This work |
| | | | 200 | 775.2 | |

To further investigate the chemical reaction kinetics of the $Fe_2O_3$@3DG-150 during the charging and discharging process, the CV curves at different scan rates were tested (Figure 6a). They show a similar electrochemical behavior and the redox peaks gradually broaden with increasing scan rate. The electrode electrochemical storage process can be described as a surface capacitance process and a diffusion-controlled insertion process. In order to measure the contribution of the two mechanisms, the relationship between the scan rate ($v$) and the peak current ($i$) can be described according to the following equation: $log(|i|) = blog(v) + log(a)$. The b-value can be characterized from the slope of the linear fit between $log(v)$ and $log(|i|)$ [35,49]. In general, electrodes exhibit diffusion control at a b-value of 0.5 and surface capacitance control kinetic processes at a b value of 1. The b-value ranging from 0.5 to 1.0 indicates the mixing kinetic process during the discharge-charge reaction. The results of fitting the b-values according to the peaks of the cathodic and anodic processes are shown in Figure 6b. The b-value for the cathodic peak is 0.56, indicating that the deintercalation of $Li^+$ is mainly the diffusion-controlled process. The b-value for the anodic peak exhibits 0.70, implying that the intercalation of $Li^+$ is a combination of the capacitive and diffusion-controlled process [42,49]. This difference in b-values is caused by some lithium remains in the material after discharge so that $Fe_2O_3$ is not fully

recovered. The charge-discharge processes are accompanied by pseudo-capacitance and slight crystallographic phase change [42]. The 1D structures are capable of limiting the $Fe_2O_3$ phase transition to the axial direction, making the pseudo-capacitance process highly reversible, which leads to its beneficial cycling capability.

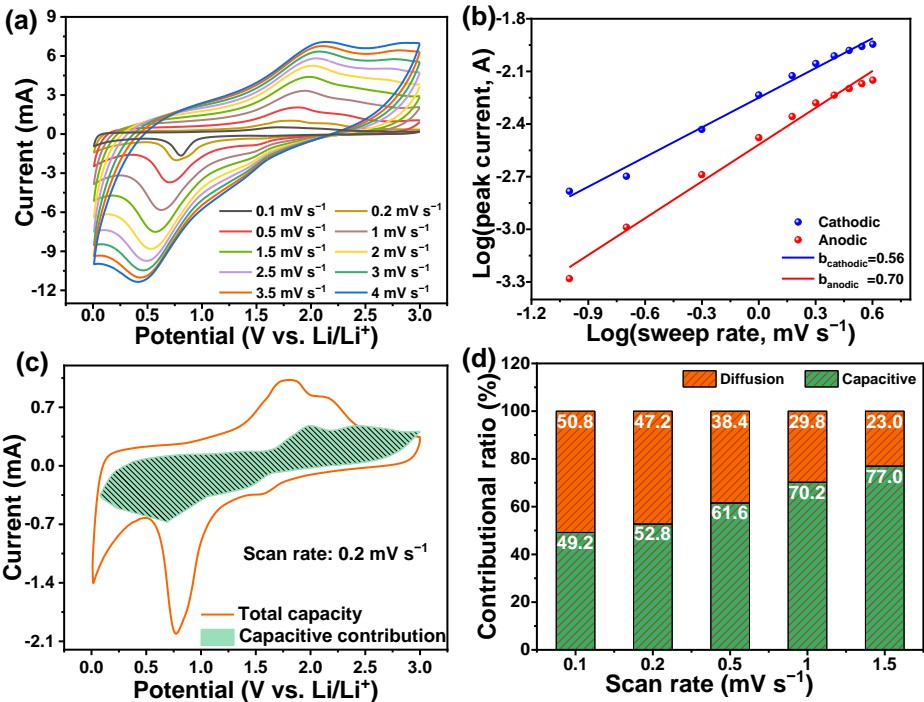

**Figure 6.** (**a**) CV curves $Fe_2O_3$@3DG-150 at various scan rates. (**b**) The relationship between log (peak current) and log (sweep rate) in the cathodic and anodic processes. (**c**) CV curve that separates into capacitive contribution (green region) and total curve at a scan rate of 0.2 mV s$^{-1}$. (**d**) The percentages of capacitive-effect contribution and diffusion-controlled contribution at different scan rates.

Then, in order to further identify the contribution of the total capacitance, the relationship between the current response (*i*) at a fixed potential (*V*) can be expressed according to the equation: $i(V) = k_1\nu + k_2\nu^{1/2}$, where $k_1\nu$ and $k_2\nu^{1/2}$ represent the surface capacitance contribution process and diffusion-controlled process respectively. The CV curve (Figure 6c) at 0.2 mV s$^{-1}$ is calculated using the above relationship, where the solid part indicates the capacitance contribution region, the orange curves indicate the total capacitance contribution region, and the latter minus the former represents the diffusion control region. According to the results, 52.8% of the total capacitance is attributed to capacitive effect, while the remaining part is due to diffusion effect. Figure 6d shows the normalized contribution ratio of $Fe_2O_3$@3DG-150 at various scan rates. These results confirm the improved electrochemical properties of the $Fe_2O_3$@3DG-150 can probably be attributed to the 1D structure of $Fe_2O_3$, which is uniformly distributed and pierced in the 3DG framework, creating a stable 1D-3D interpenetrating structure, effectually relieving the volume expansion. Furthermore, the distinctive 1D nanostructure of $Fe_2O_3$ offers numerous channels for electrolyte ingress, thereby enhances ion transmission. The self-assembled 3DG can enhance the electric conductivity of the integral electrode.

The Nyquist plots of $Fe_2O_3$@3DG-150 and $Fe_2O_3$-150 were shown in Figure 6. The Nyquist plots consist of two semicircles present in the high and medium frequency ranges, and a linear section at the low-frequency end. The semicircle at the high frequency ranges is related to the resistance of the solid-state interface layer on the surface of the electrode ($R_s$) [17]. The semicircle at medium frequencies denotes the charge transfer resistance ($R_{ct}$) between the electrolyte and electrode, while the slope of the straight line in the low frequency region indicates the diffusion resistance of Li$^+$ in the electrode material [50].

Through the equivalent circuit diagram in the inset of Figure 7 can be calculated the $R_s$ and $R_{ct}$ [28]. The $R_s$ values of $Fe_2O_3$@3DG-150 and $Fe_2O_3$-150 are 1.7 and 2.2 $\Omega$, respectively, while their $R_{ct}$ values are 60 and 141.2 $\Omega$, respectively. These results further confirm the effective linkage between 1D-$Fe_2O_3$ and 3DG resulting in an enhanced electronic conductivity and $Li^+$ diffusion rate.

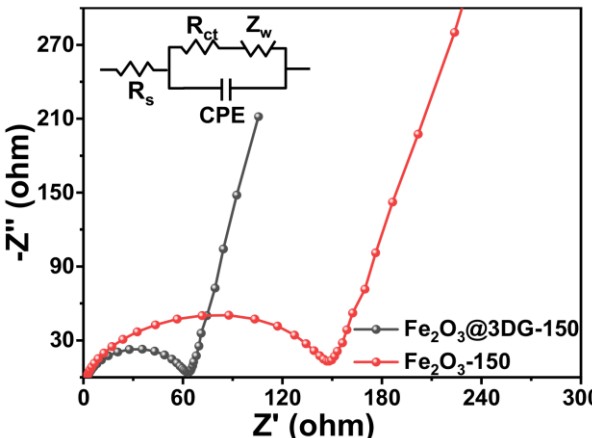

**Figure 7.** EIS plots of the $Fe_2O_3$@3DG-150 and $Fe_2O_3$-150. Inset is the fitted equivalent circuit.

## 3. Materials and Methods

### 3.1. Preparation of $Fe_2O_3$@3DG Composite

GO was prepared using the modified Hummer's method [51]. The $Fe_2O_3$@3DG composites were synthesized according to the following procedure. Initially, 0.946 g $FeCl_3·9H_2O$ and 0.497 g $Na_2SO_4$ were dissolved in 47 mL distilled water under continuous magnetic stirring to form light yellow mixed solution. Then, 23 mL GO suspension (6.6 g $L^{-1}$) was added to the mixed solution followed by continuous ultrasonic stirring. The above mixture was transferred into a 100 mL hydrothermal reactor and kept at 150 °C for 6 h. The obtained product was collected after centrifugal washing and freeze-drying. Finally, the centrifuged and freeze-dried precipitates were annealing in Ar atmosphere at 450 °C for 2 h to obtain $Fe_2O_3$3DG-150 composite materials. For comparison, the reference samples were changed to a hydrothermal reaction temperature of 120 °C and 180 °C for synthesis and named $Fe_2O_3$@3DG-120 and $Fe_2O_3$@3DG-180, respectively. Meanwhile, the hydrothermal solution of $FeCl_3$ and $Na_2SO_4$ without the addition of GO was used for the experiment at 150 °C. The resulting sample was named $Fe_2O_3$-150.

### 3.2. Physical Characterization

Microscopic structure and morphology of the samples were observed by scanning electron microscopy (SEM, Nova NanoSEM400, CA, USA) and transmission electron microscopy (TEM, JEM-2000 UHR SETM/EDS, Tokyo, Japan) with their accompanying energy dispersive X-ray spectroscopy (EDX). An Xpert Pro MPD diffractometer (XRD, Cu K$\alpha$ source, Nalytical, Eindhoven, The Netherlands) was used to analyze the physical phase and crystalline structures of the samples. X-ray photoelectron spectroscopy (XPS, Al K$\alpha$ source, AXIS SUPRA+, Kyoto, Japan) was performed to obtain the chemical valence states of Fe, O and C elements in the samples. Thermogravimetric analysis (TGA, Netzsch, STA449, Selb, Germany) was carried out to obtain the 3DG content in the samples. An $N_2$ adsorption/desorption apparatus (ASAP 2460, Micromeritics, GA, USA) was used to manifest porous structure, and the samples were degassed at 200 °C under vacuum for 6 h before measurement.

### 3.3. Electrochemical Measurements

The test electrodes were prepared by forming a slurry of active materials, acetylene black and polyvinylidene fluoride (PVDF) in an 8:1:1 weight ratio with N-methyl-2-

pyrrolidone (NMP) as the solvent, which was then coated onto a copper foil. The mass of active materials loaded was about 1.5 mg. Pure lithium foil, Celgard 2400 microporous polypropylene membrane were counter electrode and separator, respectively. The electrolyte was 1 M $LiPF_6$ dissolved in ethylene carbonate and Diethyl carbonate (1:1 vol). All coin cells (CR 2032) were manually assembled in a high purity argon-filled glovebox. Cyclic voltammetry (CV) measurements and electrochemical impedance spectroscopy (EIS) measurements were implemented using a VMP3 (Bio-Logic, Chatou, France) electrochemical workstation. The EIS test frequency was between 0.01 Hz to 100 kHz with 5 mV amplitude. The galvanostatic charge and discharge tests were conducted using a Neware automatic battery cycler in the voltage range of 0.01 and 3.0 V.

## 4. Conclusions

In summary, we fabricated successfully $Fe_2O_3$@3DG nanocomposites with various nanostructures by temperature and annealing strategy. We explored the temperature-driven crystallographic morphology evolution and achieved the controlled synthesis of $Fe_2O_3$ with different structures (nanocubes,1D-nanorods and ellipsoidal crystals). The results show that the 1D $Fe_2O_3$ synthesized at 150 °C can easily pierce into the 3DG framework to form a stable 1D-3D interpenetrating structure. This structure possesses graded porous features that create a large surface area (129.4 $m^2$ $g^{-1}$) to provide abundant lithium storage sites while providing unparalleled structural stability and capacity reversibility. Moreover, the 1D and 3D interpenetrating hybridized structure delivers good stability and cyclic reversibility. When served as anode materials for LIBs, the 1D-3D $Fe_2O_3$@3DG exhibits an initial discharge specific capacity of 1451 mAh $g^{-1}$ at 0.1 A $g^{-1}$, and a high reversible specific capacity of 775 mAh $g^{-1}$ after 200 cycles. The ideas of structural design presented in this paper have the potential to optimize the performance of other TMOs materials and enable their application in commercial LIBs.

**Author Contributions:** S.Z., J.X. and R.L. wrote the original draft of the paper. S.Z. and L.Y. designed the experiments, synthesized the materials, prepared the electrodes, performed physical characterization, the electrochemical characterization, and analyzed the data. R.L. wrote the review and edit. R.L. and Y.Z. provided supervision and funding acquisition. All authors have read and agreed to the published version of the manuscript.

**Funding:** This research was funded by the National Natural Science Foundation of China (no. 51802233).

**Data Availability Statement:** Not applicable.

**Conflicts of Interest:** The authors declare no conflict of interest.

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
