# Peer review of "Temperature-Driven Synthesis of 1D Fe2O3@3D Graphene Composite Applies as Anode of Lithium-Ion Batteries"

_inorganics, doi:10.3390/inorganics11050211_

Round 1
Reviewer 1 Report
In the work authors reported hydrothermal and annealing methods of Fe2O3@3DG synthesis with the potential to obtain composites for the design of advanced anode materials for lithium-ion batteries. The obtained materials were characterized by SEM; TEM, XPS, XRD, and TGA.
I recommend accepting for publishing after a major revision. In general, the quality of the manuscript makes it really difficult to assess the exact value of the conclusions. Following are some comments for the authors to improve the manuscript:
1) Missing data on the characterization of samples Fe2O3@3DG-120 and -180. It is necessary, in addition to the SEM and TGA results, to show the results of the comparative characterization of the tested materials, on the basis of which the authors draw conclusions about the electrochemical properties of the examined materials. If due to the length of the manuscript, all characterization results could not be presented in the manuscript they should be represented in the supporting file.
2) It is important to have BET or ECSA results for a better interpretation of the electrochemical properties of examined anode materials.
3) Additionally, the electrochemical study of the pseudocapacitive behavior of the tested materials would give a significant contribution to the quality of conclusions, since it is known that carbon material/metal oxide structures have two modes of charge storage, electric double-layer capacitance (EDLC) and pseudo-capacitance, which can be decoupled using Dunn analysis. doi:10.1039/c3ee44164d.
4) The conclusion should be written more specifically in relation to the synthesis conditions and the most significant results
Author Response
Thanks for your comments. We have answer them point-by- point. Please see the attachment.

Reviewer 2 Report
The results presented by the authors are interesting, but some corrections are necessary before the acceptance:
1. The authors must read the manuscript carefully to correct typos.
2. The authors must present the values of energy system for each graphene composite studied in this work.
3. The authors must explain which are the redox centers for the charge storage.
4. Add the chemical reactions for the charge storage and explain them.
5. The authors must add a table to present values of mAh g-1 of previous similar composites and compare with the results of this research.
6. Which are the novelties and contributions of this research? It is not clear. The novelties of the graphene composites studied here must be added in the introduction section.
7. Which are the advantages of your composites with respect to previous ones?
8. It is not clear how the morphology is related with the peformance of the electrode. Please explain clearly in the manuscript.
9. Authors must add BET measurements to correlate the surfacea area with the performance.
10. The background of the manuscript must be improved. There are recent reports about the use of graphene composites for supercapacitors. Those devices have a battery-like component and their discharge is slow. Thus, the following references must be discusssed/added in the introduction section to improve the background:
a) Synthetic Metals, Volume 264, June 2020, 116384
https://doi.org/10.1016/j.synthmet.2020.116384
The references above will be useful to explain the potential of graphene composites as electrodes in supercapacitors.
The authors must improve the english style and grammar.
Author Response

(The authors gave the same response as above.)

Round 2
Reviewer 1 Report
The article can be accepted for publication.
Author Response
Dear Reviewer:
Thank you for providing valuable suggestions. We have made revisions as suggested. Changes are highlighted by red color in the Revised Manuscript.
It will be greatly appreciated if the revised version can be acceptable for publication in Inorganics.
_____ Thank you for your approval of our revised manuscript. Wish you all the best.
Reviewer 2 Report
The article can be accepted for publication.
Author Response
Dear Reviewer:
Thank you for providing valuable suggestions. We have made revisions as suggested. Changes are highlighted by red color in the Revised Manuscript.
It will be greatly appreciated if the revised version can be acceptable for publication in Inorganics.
——Thank you for your approval of our revised manuscript. Wish you all the best.